

# Bidirectional encoder representations from transformers and deep learning model for analyzing smartphone-related tweets

Sudheesh R[1], Muhammad Mujahid[2], Furqan Rustam[3], Bhargav Mallampati[4], Venkata Chunduri[5], Isabel de la Torre Díez[6] and Imran Ashraf[7]

[1] Kodiyattu Veedu, Kollam, Valakom, India

[2] Department of Computer Science, Khwaja Fareed University of Engineering and Information Technology, Rahim Yar Khan, Pakistan

[3] School of Computer Science, University College Dublin, Dublin, Ireland

[4] College of Engineering, University of North Texas, Denton, TX, United States of America

[5] Indiana State University, Terre Haute, IN, United States of America

[6] Department of Signal Theory, Communications and Telematics Engineering, University of Valladolid, Valladolid, Spain

[7] Information and Communication Engineering, Yeungnam University, Gyeongsan, Republic of Korea

Corresponding author
Imran Ashraf, imranashraf@ynu.ac.kr

## ABSTRACT

Nearly six billion people globally use smartphones, and reviews about smartphones provide useful feedback concerning important functions, unique characteristics, etc. Social media platforms like Twitter contain a large number of such reviews containing feedback from customers. Conventional methods of analyzing consumer feedback such as business surveys or questionnaires and focus groups demand a tremendous amount of time and resources, however, Twitter's reviews are unstructured and manual analysis is laborious and time-consuming. Machine learning and deep learning approaches have been applied for sentiment analysis, but classification accuracy is low. This study utilizes a transformer-based BERT model with the appropriate preprocessing pipeline to obtain higher classification accuracy. Tweets extracted using Tweepy SNS scrapper are used for experiments, while fine-tuned machine and deep learning models are also employed. Experimental results demonstrate that the proposed approach can obtain a 99% classification accuracy for three sentiments.

## INTRODUCTION

Social media enables businesses to interact with prospective customers in a timely, informative, and cost-effective manner. Currently, a huge number of executives are interested in social communication networks. Users of social media can participate in discussions, share information, and publish content related to products and services (*Kaplan & Haenlein, 2010*). Social media include blogs, microblogs, social networking websites, image, and video hosting services, instant messaging, and posting

content from a large number of users. Through social media, we may communicate with the rest of society and express our thoughts, ideas, and opinions. Due to its widespread acceptance as an efficient communication resource, social media especially Twitter with nearly 200 million users is becoming more popular. Twitter is a fast-growing instant messaging network where users "tweet" to communicate information. These tweets offer people's opinions and knowledge about social and business problems (*Pak & Paroubek, 2010*).

Nowadays, most companies use a variety of techniques to improve their products. Companies mostly use customer exposure as a strategy to understand customers' opinions. Feedback surveys, systematic forms, ratings, and remote monitoring are typical ways to get feedback from customers. Using the information acquired from these comments, business companies can improve quality of products and services (*Liu, 2012*). The smartphone industry has been expanding dramatically, not just through traditional sales but also through internet sales. Customers search for the best phone and its features through online platforms and share information on social media (*Silver, Huang & Taylor, 2019*). Smartphones are becoming ever more essential in ordinary routine, and they offer a huge variety of platforms for information, communication, education, and entertainment (*Barkhuus & Polichar, 2011*). Smartphones typically feature touch screens, mobile Internet connectivity through WiFi or mobile networks, and the ability to install applications. Twitter is more valuable for companies as users express their opinions and sentiments in the form of short and long texts on any product (*Jansen et al., 2009*). Businesses and organizations struggle to gather tweets and analyze the containing sentiments. Tweets may be quickly assessed and classified into positive or negative emotions using automated sentiment analysis.

Several works have been presented that utilize machine and deep learning approaches for automatic sentiment classification. Parts of speech (PoS), lexicon-based approaches, and TextBlob are privileged techniques in this regard. For example, sentiments regarding reviews of smartphones are classified using support vector machine (SVM) in *Kumari, Sharma & Soni (2017)* and reported a 91% accuracy. Similarly, *Krishnan, Sudheep & Santhanakrishnan (2017)* used only a novel lexicon-based method to perform sentiment analysis on tweets for different mobile phones including the iPhone, Lenovo, Motorola, Nexus, and Samsung. *Gurumoorthy & Suresh (2022)* evaluated the sentiments related to popular smartphone products involving the natural language processing (NLP) toolkit and TextBlob. Such approaches have several restrictions. Machine learning and deep learning approaches are no extensively studied. A few studies focused on tweets labeling regarding sentiments while other deployed machine and deep learning models for sentiment classification. In addition, preprocessing techniques are not utilized effectively and their impact on classification accuracy is not extensively studied. Models are not evaluated in terms of computation cost. This study focuses on the above-mentioned limitations and makes the following contributions

1. A large number of tweets are extracted about smart phones from Twitter for brands like Apple, Samsung, and Xiaomi. Since extracted tweets are unstructured, preprocessing is performed to remove unnecessary and redundant data. Preprocessing being important

to enhance the efficacy of models, the impact of preprocessing is analyzed regarding models' performance and time complexity.

2. TextBlob is utilized to extract the polarity and subjectivity from tweets related to smartphone brands and label them as positive, negative, or neutral. The bag of words (BoW), term frequency-inverse document frequency and Word2Vec feature engineering techniques are used to extract relevant features.

3. A transformer-based bidirectional encoder representations from transformers (BERT) model is proposed to accurately classify the sentiments. The effectiveness and reliability of the proposed model are checked against other models. Furthermore, the proposed model's robustness is evaluated on the additional dataset.

4. Different machine learning models like logistic regression (LR), random forest (RF), K nearest neighbor (KNN), SVM, stochastic gradient descent (SGD), decision tree (DT), extra tree classifier (ETC) and gradient boosting machine (GBM) are fine-tuned to obtain the best results and compared their performance the proposed BERT model.

This article is further divided into four sections. Section 2 presents the details of the literature review relevant to the current problem. Section 3 contains the materials and proposed approach in which we briefly discuss dataset information, preprocessing, machine and deep learning models, and their architecture. The results and discussion are summarized in Section 4. The conclusion is given in Section 5.

## RELATED WORK

Sentiment classification is an important research area in NLP and has been investigated widely during the past few years. For example, *Naramula & Kalaivania (2021)* collected tweets regarding iPhone and Samsung using the natural language toolkit (NLTK) and utilized machine learning models including RF, KNN, and SVM to classify tweets. Similarly, *Jagdale, Shirsat & Deshmukh (2019)* product review dataset for sentiment analysis using the lexicon-based approach with machine learning models. However, results from machine learning are not compared with other models, and the dataset was also limited. Combining lexicon-based and SVM, study (*Chamlertwat et al., 2012*) utilized microblog sentiment analysis to assist smartphone manufacturers. The study determined that some Apple customers tweet about necessary defects of mobile devices.

Along the same direction, *Ray & Chakrabarti (2017)* used lexicon-based, aspect-level, and document-level sentiment analysis on product reviews collected from Twitter. Over 3,000 tweets are collected and preprocessed for experiments. The classification is done using a lexicon dictionary-based method and emotions are detected. *Fang & Zhan (2015)* performed sentence-level and review-level sentiment analysis on Amazon product reviews using both manually labeled tweets and automatically labeled tweets. For sentiment classification, three machine learning models naive Bayes (NB), RF and SVM are utilized. To conduct an accurate analysis, the tweets' punctuation, misspellings, and slang words are eliminated. Following that, a feature vector is developed using pertinent features. The classification of tweets into positive and negative classifications is completed using various classifiers (*Neethu & Rajasree, 2013*).

Twitter tweets relating to mobile phones are collected in *Driyani & Walter Jeyakumar (2021)* for sentiment analysis. The SVM model is used for sentiment classification on three different variations of the dataset. SVM is evaluated using different kernels and four different cross-validation approaches are also utilized. Results show that the radial-based function (RBF) kernel performs better than any other kernel. In addition, the performance of SVM is reported to decrease while increasing the dataset size. In *Singla, Randhawa & Jain (2017)*, positive and negative moods in reviews regarding mobile phones are collected from Amazon.com and analyzed. Reviews are categorized using a variety of models including NB, SVM, and DT. Results report an 81.87% accuracy with SVM. *Chawla, Dubey & Rana (2017)* carried out a sentiment analysis on text from smartphone reviews. Of the NB and SVM models, SVM is reported to have good results.

Amazon mobile phone reviews are utilized for sentiment analysis in *Dhabekar & Patil (2021)*. The tweets are processed and labeled using Vader-Analyzer. The long short-term memory (LSTM) model with one embedding layer including 256,800 parameters, an LSTM layer containing 256,800 parameters, and a dense classification layer containing 394 parameters are used for experiments. The proposed LSTM model achieved a 93% accuracy. Similarly, *Onan (2019)* analyzed the sentiment of product reviews using a convolutional neural network (CNN) model with deep learning models. Deep learning and the word embedding techniques fastText, GloVe, and Word2Vec are employed to classify sentiments. Additionally, the authors compare the proposed deep learning model to conventional machine learning models and reported promising results.

*Iqbal et al. (2022)* used an LSTM model with various combinations of layers for sentiment categorization on five different product review datasets obtained from Twitter and Amazon. The study used a combination of LSTM and CNN in sentiment analysis of tweets (*Umer et al., 2021*). Moreover, the efficacy of Word2Vec and term frequency-inverse document frequency techniques is evaluated. The study demonstrates that deep learning models outperform machine learning. Table 1 presents a comparative analysis of the discussed research works.

The authors (*Supriyadi & Sibaroni, 2023*) used Twitter tweets related to the Xiaomi smartphone for sentiment analysis with different aspects like camera, random access memory (RAM), and screen size. The authors first preprocessed the tweets and then divided the cleaned dataset into training and testing, separately. To analyze the opinions of the public towards different smartphone aspects, they adopted BERT and IndoBERT models for analysis. The proposed IndoBERT model achieved 90% accuracy. The authors achieved 78% positive sentiments with battery aspects, 76% on RAM aspects, and 68% on camera aspects. With battery quality aspects, IndoBERT achieved 18% negative sentiments. *Sally (2023)* extracts reviews for the Samsumg Galaxy S21, iPhone 13, and Google Pixel 6. The downloaded data contains text, numbers, and emojis. The reviews are in different languages, but for sentiment analysis, they utilized reviews in English only. The numbers and emojis are removed because they contain no meaningful data and only textual data are considered. The textual data were mostly labeled using the VADER technique into positive, negative, and neutral tweets. The feature extraction is performed using BoW. To classify the sentiments, the authors employed SVM, NB, and DT classifiers. Out of these

**Table 1  Summary of related work.**

| Reference | Model | Datasets | Results |
|---|---|---|---|
| *Jagdale, Shirsat & Deshmukh (2019)* | SVM | Amazon product reviews | 92.85% accuracy, 91.64 precision, 95.64 f1-score |
| *Chamlertwat et al. (2012)* | Lexicon based + machine learning approach | Smartphone brands tweets | The authors only extract positive or negative sentiments. |
| *Fang & Zhan (2015)* | SVM, NB, RF | Amazon product reviews | The Naïve Bayes and SVM achieved same f1 score on review level sentiment classification. |
| *Driyani & Walter Jeyakumar (2021)* | SVM with RBF kernel | Apple iPhone reviews | SVM with the RBF kernel, and only 18,000 reviews achieved 91.87% accuracy. |
| *Chawla, Dubey & Rana (2017)* | SVM and NB | Smartphone related reviews | The naive Bayes model achieved 40% accuracy, and the SVM achieved 90% accuracy on smartphone reviews. This study does not extract important features from the data. The deep learning experiments are also missing. |
| *Dhabekar & Patil (2021)* | LSTM | Amazon products | 93% accuracy, 93 precision , 93 recall, 92 f1 score |
| *Iqbal et al. (2022)* | LSTM | Amazon products | LSTM with different layers achieved better results on amazon food reviews, smartphone accessories, Yelp, amazon products, and IMDB tweets datasets. |
| This study | BERT | SmartPhone | The proposed BERT model achieved 99.3% accuracy by utilizing preprocessing techniques, and 98.4% accuracy without applying preprocessing to the smartphone-related tweets dataset. |

classifiers, SVM attained 78% accuracy. The authors did not adopt any deep transformers that learn complex representations of data and achieve poor results. Also, this study does not compare the classifiers with other methods to validate the results.

*Yuhan & Huiping (2023)* used aspect-level sentiment analysis of smartphone-related reviews using the context window self-attention (CWSA) model. On the Chinese dataset, the authors achieved an F1 score of 89.6%. They used a limited dataset for aspect-level analysis. Similarly, in *Baydogan & Alatas (2022)*, the authors used NLP techniques to gather two tweet datasets related to hate speech detection. They employed BoW and TF-IDF, two important techniques, to extract features from the datasets. They used ten ML and DL models for sentiment classification. Results proved that recurrent neural network model performs best with both datasets, with an accuracy of 78% and 90%, respectively. In another study (*Baydogan & Alatas, 2021a*), the authors utilized a hate speech dataset for sentiment classification. They used three feature extraction techniques including BoW, TF-IDF, and Word2Vec, for extracting features. Ant lion optimizations (ALO) and moth flame optimization (MFO) methods were also utilized. Results indicate that ALO and MFO methods perform with 92% and 90% accuracy, respectively, compared to machine learning. *Baydogan & Alatas (2021b)* collected 12000 unlabeled tweets and performed sentiment analysis using NLP for preprocessing and labeling the tweets. A machine learning approach is employed for classification. A spider optimization algorithm was developed that performed best compared to machine learning models and obtained an 86% accuracy.

## PROPOSED METHODOLOGY

The proposed methodology for performing sentiment analysis on smartphone brands is described in this section. Figure 1 shows the proposed workflow diagram for sentiment classification. First, the tweets dataset is extracted from Twitter for the top three smartphone brands.

It is followed by the cleaning process where a range of preprocessing steps is used. Tweets are then classified as positive, negative, and neutral using the TextBlob technique. After this, important features are extracted from the cleaned text using the count vectorizer approach. Finally, data is split into train and test sets. In addition to the proposed BERT model, a number of machine learning and deep learning models are used for sentiment classification. The details of the steps involved in the workflow are briefly described in subsequent sections.

### Dataset information

The dataset is extracted from Twitter for October 2022 to November 2022 using the tweepy SNS scrapper and the query "Apple phone, smartphone, Samsung smartphone, Xiaomi smartphone" is used. Using the search scrapper, 33,383 unstructured tweets containing punctuation, stopwords, uniform resource locator (URLs), tags, usernames, and emoticons are collected. After removing the punctuation, null and duplicate values from the tweets, a total of 32,420 tweets are used for experiments. The Twitter dataset includes the date, user name, location, and tweet text; a few sample tweets are given in Table 2.

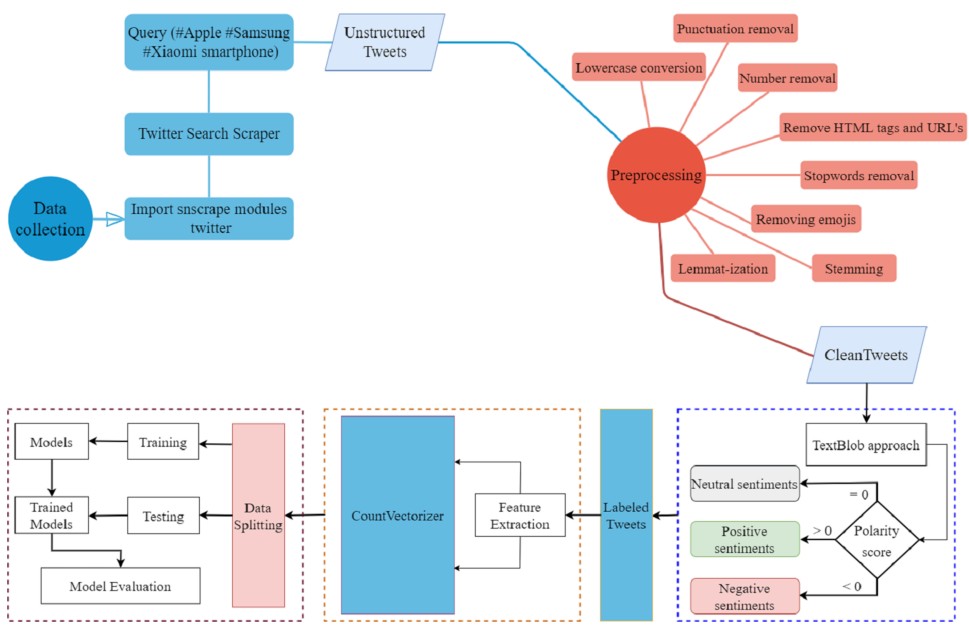

**Figure 1** Work flow diagram of the proposed approach.

**Table 2** Sample tweets from the collected dataset.

| Date | User | Location | Tweets |
|------|------|----------|--------|
| 2022-10-30 | StrayTurtle | California, USA | Think Apple would every bring back the Blackberry design? I bet people would go for it. Physical keyboard on the bottom half of an iPhone for a flip-smartphone or the screen slides over the keyboard. Had to Google it, #iPhone # Apple @Apple |
| 2022-10-28 | pickles769899 | Columbus, GA | @SamsungMobile This I the 4th time my phone has shut off to update. I just postponed it 1 hr ago! Really making me think of switching back to the IPhone. I turn my phone off every night and it could do it then. Not sure if it's just me or happens to othe |
| 2022-11-01 | Abhi_Banarasi | India | Reading about the power of expected 200 MP on @SamsungMobile S23 Ultra and it's quite impressive. Pretty excited for that. #s23ultraleaks |

Besides the collected dataset, an additional dataset is used for validation of the proposed approach. The second dataset is obtained from Kaggle and contains a total of 10K cryptocurrency-related tweets. The dataset contains tweets that were collected from January 1, 2021, to November 2022.

## Text preprocessing

Preprocessing is a technique by which the unstructured data are transformed into a comprehensible and logical format (*Vijayarani, Ilamathi & Nithya, 2015*). Preprocessing is primarily used to improve the quality of text data by reducing its quantity so that the machine can identify important patterns.

A machine learning model will also have a higher level of accuracy since the machine can learn from the data more accurately. Initial processing of data is performed in preparation for further analysis or processing. Several procedures are followed to prepare data for models' training. Following steps are performed in preprocessing, as shown in Table 3.

## TextBlob

TextBlob is an important Python library for preprocessing textual data (*Bonta & Janardhan, 2019*) and is used for sentiment analysis, noun-phrase extraction, classification, and translation. Sentiment analysis can help readers in detecting the moods and views expressed in tweets, as well as other vital information. TextBlob provides the polarity and subjectivity of a sentence. The polarity range is $[-1, 1]$, where $[-1]$ represents a negative mood and $[1]$ represents a positive mood. A sentence's polarity is modified by the use of negative words. TextBlob's semantic labels enable fine-grained analysis. Subjectivity falls in the range of 0 to 1. Subjectivity refers to the amount of personal opinion and factual information in a text. Due to the text's increased subjectivity, it contains more personal opinions than objective information. TextBlob has one extra context called intensity. TextBlob uses the intensity to calculate subjectivity. A word's intensity determines whether it changes the subsequent word. Textblob offers a wide range of features for showing particular textual data attributes. Table 4 shows the distribution of sentiments extracted using the TextBlob approach.

## Feature extraction

The count vectorizer, or 'bag of words' is a natural language processing approach for extracting features from textual data. It is a simple feature extraction approach, yet produces good results. Textual data contain unstructured and chaotic information, whereas models require structured, well-defined, fixed-length information. Count vectorizer turns textual input into number vectors, as machine learning requires numeric data. A bag of words from two preprocessed sentences is presented in Table 5.

After feature extraction, the dataset is split into a train and test set with a ratio of 0.8 to 0.2 for model training and testing, respectively.

## Machine learning models

Machine learning enables automated sentiment analysis and has been widely used recently for sentiment analysis (*Mujahid et al., 2021*), image identification (*Wang, Fan & Wang, 2021*), object detection (*Ramík et al., 2014*), information retrieval, and so on. Nine of the most well-known machine learning models are used in this sentiment analysis study with hyperparameters tuning. The values for fine-tuned hyperparameters are shown in Table 6.

### *Support vector machine*

SVM is mostly used in NLP for classification and regression tasks (*Ahmad, Aftab & Ali, 2017*). SVM uses a hyperplane to distinguish between the classes. The effectiveness of SVM increases as the number of dimension spaces increases. This model performs poorly on large datasets and requires a larger training time. SVM creates support vectors, works well on small datasets, and demands limited resources. In situations when there is a significant

**Table 3  Text preprocessing steps used in this study.**

**Steps**

**Step 1 —Lowercase conversion**

Due to the case-sensitivity of models, conversion to lowercase is essential. The model treats "SMARTPHONE" and "smartphone" as two separate words if conversion is not performed. Sample tweets before and after conversion are given here;

**Before Lowercase Conversion**

Sentence 1: Apple will remove physical Buttons from iPhone 15 Pro Smartphone

Sentence 2: Samsung Mobile, the new update was unnecessary. The notifications look bad. https://t.co/L6nzYpB8Oy

**After Lowercase Conversion**

Sentence 1: apple will remove physical buttons from iPhone 15 pro smartphone

Sentence 2: samsung mobile, the new update was unnecessary. the notifications look bad. https://t.co/L6nzYpB8Oy

**Step 2 —Removal of numbers**

Numerical values are removed to improve the model's training and reduce computational complexity. Text data typically consists of quantitative values such as digits, which are of little relevance for decision-making processes. As a consequence, the numerical values provide a significant challenge to the algorithm when it attempts to extract features from the texts. In the majority of instances, the classification of data does not include the use of values including numbers (*Anandarajan, Hill & Nolan, 2019*). When dealing with textual data or reviews that are not concerned with numbers, it is needed to remove them. Sample data before and after numbers removal are given here;

**Before number removal**

Sentence 1: apple will remove physical buttons from iphone 15 pro smartphone

Sentence 2: samsung mobile, the new update was unnecessary. the notifications look bad. https://t.co/L6nzYpB8Oy

**After number removal**

Sentence 1: apple will remove physical buttons from iphone pro smartphone

Sentence 2: samsung mobile, the new update was unnecessary. the notifications look bad. https://t.co/L6nzYpB8Oy

**Step 3 —Punctuation removal**

The third step of data preprocessing is punctuation removal which aims to remove the punctuation from the data. Punctuations are removed from data because they do not contribute to the learning of a machine learning model. Also, it reduces the machine's ability to differentiate between other characters and punctuation. The following samples illustrate the punctuation removal process;

**Before Punctuation removal**

Sentence 1: apple will remove physical buttons from iphone pro smartphone

Sentence 2: samsung mobile, the new update was unnecessary. the notifications look bad. https://t.co/L6nzYpB8Oy

**After Punctuation removal**

Sentence 1: apple will remove physical buttons from iphone pro smartphone

Sentence 2: samsung mobile the new update was unnecessary the notifications look bad https://t.co/L6nzYpB8Oy

**Table 3** (*continued*)

**Step 4 —Stopwords removal**

Preprocessing involves deleting non-classifiable objects from the data set. Stop words clarify the meaning for humans, but for machine learning models they do not add any value and are thus removed (*Pradana & Hayaty, 2019*). Stopwords include "is", "am", "I", "the", "to", "are", "that", "they," etc.

**Before Stopwords removal**

Sentence 1: apple will remove physical buttons from iphone pro smartphone

Sentence 2: samsung mobile the new update was unnecessary the notifications look bad https://t.co/L6nzYpB8Oy

**After Stopwords removal**

Sentence 1: apple remove physical buttons iphone pro smartphone

Sentence 2: samsung mobile new update unnecessary notifications look bad https://t.co/L6nzYpB8Oy

**Step 5 —Removing emojis**

Emojis should be removed from the tweets to attain the best results. Tweets may also feature emojis which are the pictorial presentation of different moods like anger, sadness, happiness, etc. Sample tweets before and after emojis removal are given below;

**Before removing emoji**

Sentence 1: apple remove physical buttons iphone pro smartphone

Sentence 2: samsung mobile new update unnecessary notifications look bad https://t.co/L6nzYpB8Oy

**After removing emoji**

Sentence 1: apple remove physical buttons iphone pro smartphone

Sentence 2: samsung mobile new update unnecessary notifications look bad https://t.co/L6nzYpB8Oy

**Step 6 —URL'S and HTML tags**

The URLs and HTML tags do not provide any useful information for the model's training and are removed from the tweets to enhance the performance of models. Tweets with and without such tags are given here;

**Before Conversion**

Sentence 1: apple remove physical buttons iphone pro smartphone

Sentence 2: samsung mobile new update unnecessary notifications look bad https://t.co/L6nzYpB8Oy

**After Conversion**

Sentence 1: apple remove physical buttons iphone pro smartphone

Sentence 2: samsung mobile new update unnecessary notifications look bad

**Step 7 —Stemming**

During the stemming process, the words in the data are converted back to their original form. The effectiveness of machine learning models improves by using stemming (*Pradana & Hayaty, 2019*). Use of variations of the same word like 'worry', 'worried', 'worrying', etc. increases computational complexity and reduces the model's performance. Stemming process is illustrated in the following examples;

**Before stemming**

Sentence 1: apple remove physical buttons iphone pro smartphone

Sentence 2: samsung mobile new update unnecessary notifications look bad

**After stemming**

Sentence 1: appl remov physic button iphon pro smartphon

Sentence 2: samsung mobile new updat unnecessari notifi look bad

**Table 3** (*continued*)

| Step 8 —Lemmatization |
| --- |
| Lemmatization converts all words into their base form. However, unlike stemming which removes the last characters from the words, lemmatization converts them into their original base form. Examples are given as follows; |
| **Before lemmatization** |
| Sentence 1: appl remov physic button iphon pro smartphon |
| Sentence 2: samsung mobile new updat unnecessary notify look bad |
| **After lemmatization** |
| Sentence 1: appl remov physic button iphon pro smartphon |
| Sentence 2: samsung mobile new updat unnecessari notif look bad |

**Table 4  Sentiments of three Smartphone brands by TextBlob approach.**

| Brands | Positive | Negative | Neutral |
| --- | --- | --- | --- |
| Apple | 3126 (40%) | 1035 (13%) | 3619 (47%) |
| Samsung | 6505 (32%) | 3206 (16%) | 10488 (52%) |
| Xiaomi | 2042 (43%) | 535 (11%) | 2200 (46%) |

**Table 5  Two sample sentences from the preprocessed tweets are utilized for the BoW features.**

| Sentence 1 | | | | Best smartphone offer highest custom satisfaction global | | | | | | |
| Sentence 2 | | | | Global smartphone market apple grow | | | | | | |
| | | | | Count Vectorizer Features | | | | | | |
| Sentence | best | smartphone | offer | highest | custom | satisfaction | global | market | apple | grow | Total length |
| --- | --- | --- | --- | --- | --- | --- | --- | --- | --- | --- | --- |
| Sentence 1 | 1 | 1 | 1 | 1 | 1 | 1 | 1 | 0 | 0 | 0 | 7 |
| Sentence 2 | 0 | 1 | 0 | 0 | 0 | 0 | 1 | 1 | 1 | 1 | 5 |

margin of separation among the classes, it performs with high accuracy. The disadvantage of SVM is that it cannot work well for overlapping target labels. Different kernels are used to transform the data and create an optimal hyperplane. We used a linear kernel with 3.0 cost parameters and 100 random states. Figure 2 show the SVM.

### Logistic regression

LR is the most popular model for sentiment classification (*Prabhat & Khullar, 2017*). It is easy to implement on small or large datasets and provides the results quickly and accurately. The output of this model is based on one or more independent variables. The model performs poorly when the dataset features are irrelevant and unclear. LR is easy to understand and efficient. It provides clear information about feature engineering. The LR needs careful consideration for implementing in multiclass datasets and will not perform well without feature engineering. The number of features may not be greater than the number of samples for successful results, and L1 or L2 regularization is used to avoid overfit.

**Table 6  Parameters tuning for machine learning models.**

| Models | Parameters tuning |
|--------|-------------------|
| LR | random_state=150, solver='newton-cg',multi_class='multinomial', $C = 2.0$ |
| RF | n_estimators=100, random_state=50, max_depth=150 |
| DT | random_state=150, max_depth=300 |
| ETC | n_estimators=100, random_state=150, max_depth=300 |
| SVM | kernel='linear', $C = 3.0$, random_state=100 |
| SGD | loss="hinge", penalty="l2", max_iter=5 |
| KNN | n_neighbors=3 |
| GBM | n_estimators=50, random_state=150, max_depth=200 |
| ADA | n_estimators=100, random_state=50 |

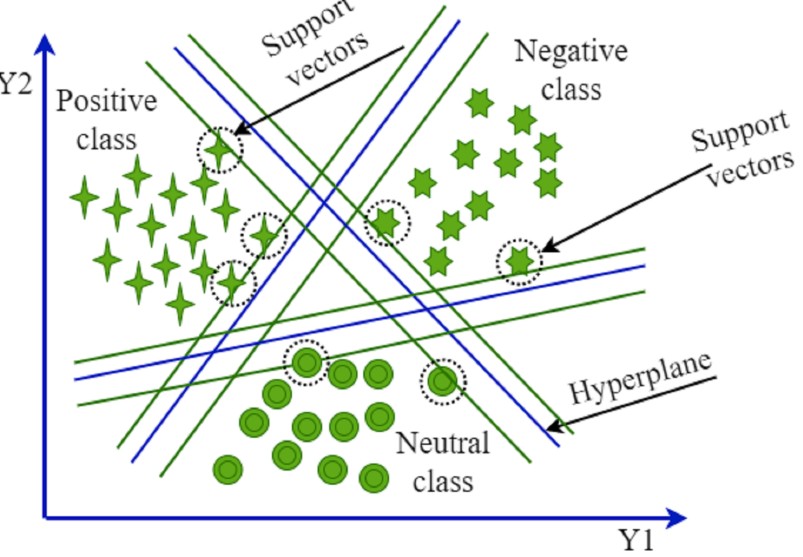

**Figure 2  Support vector machine.**

### Decision tree

A DT is a type of supervised algorithm that is hierarchically structured and composed of a root node, branches, internal nodes, and leaf nodes. The model may be applied to both numerical and categorical data, which is one of its most attractive features (*Quinlan, 1996*). The complex decision tree may result in overfitting. They cannot provide accurate predictions like RF and SVM. It is easy to understand and consider the missing values. They need feature engineering for accurate predictions. The root of a decision tree is called the initial point of decision-making. The leaf with nodes represents the final output.

### Random forest

RF is a classification method that employs ensemble learning for prediction. It generates a forest of decision trees that is more accurate than individual trees. RF generates a large

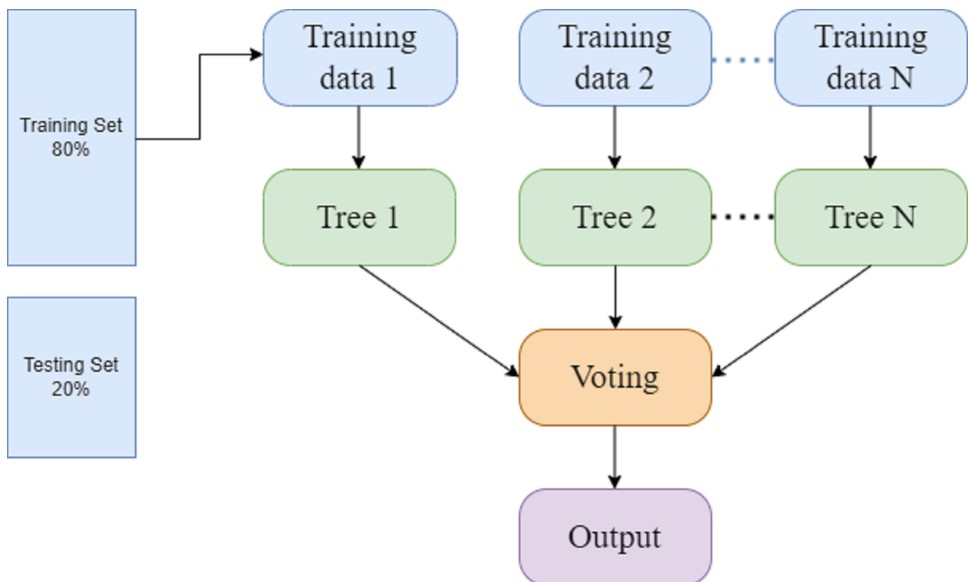

**Figure 3** **Random forest.**

number of decision trees, which increases the computation. Whereas the extra tree classifier used several decision trees, like RF, and trained on the complete dataset and may reduce bias (*Pervan & KELEŞ, 2017*). Figure 3 presents the random forest model in which 80% data is used for training and 20% for testing purpose. The different decision trees are processed through the training data. By combining various decision trees and employing a majority vote to get a final output.

$$F = mode \, n1(x), n2(x), \ldots, Nt(x) \tag{1}$$

$$F = mode \sum_{r=1}^{R} [nt(x)]. \tag{2}$$

In Eqs. (1) and (2), F represents the final prediction along with majority votes, and mode n1(x), n2(x),…, nt(x) indicates the decision trees that are used in the decision process.

### K nearest neighbor and stochastic gradient descent

SGD is a simple and efficient linear model for classification tasks (*Moh et al., 2015*). The model is trained using a maximum of 5 iterations with hinge loss and penalty L2 to achieve the best results. The SGD model supports three losses such as hinge, modified hubber, log-loss, and l1, l2, and elasticent penalties with fine tuning to get better results. The KNN (*Hota & Pathak, 2018*) algorithm uses previously obtained data to classify new data samples. It is a lazy learner and its computational complexity is high as it uses all the data for training. KNN may be used for binary and multi-class prediction. This model used training data to find the K-nearest match and employed a label for the predictions. Conventionally, Distance is calculated to find the nearest match. KNN is can be expensive

because it keeps all the training samples. The KNN has a limitation; a small number of K might result in overfitting, whereas a larger number of K results in underfitting.

$$H_t = \sqrt{(\sum_{i=1}^{m}[(K_a - U_b)])} \tag{3}$$

Equation (3) applies L2 normalization to calculate distances, with $K_a$ and $U_b$ as for prediction values at unidentified points and $H_t$ substituting for total distance.

### Extra tree classifier and AdaBoost

Extra tree, or extremely randomized tree, is a supervised ensemble learning algorithm that creates multiple DTs using the complete dataset. There is a minor difference between ETC and RF, RF chooses the best-split nodes, and ETC splits the node in a randomized fashion. According to time and computational cost, ETC is fast and efficient. We achieved the best results by increasing the number of estimators to 100 and the maximum depth to 100. ADA boost, also known as an adaptive boost, is a popular and commonly used algorithm. Weak leaners are combined to create a strong learner. The algorithm focuses on and selects samples from previous iterations that were misclassified with larger weights in each iteration. Then, using this set of weighted samples, the next weaker learner is trained. The process is repeated as many times as possible or until an acceptable degree of accuracy is achieved.

## Deep learning models

The performance of a machine learner is dependent on feature extraction and numerous studies have concentrated on developing effective feature extractors utilizing domain expertise. Deep learning algorithms have shown to be superior at analyzing and modeling complicated linguistic structures. Deep learning discovers complex information from the data without feature engineering. These models achieved state-of-the-art results for sentiment analysis. Deep learning models such as LSTM, CNN, RNN, BiLSTM, and GRU are also used for text classification. Trainable parameters of deep learning models are shown in Table 7 and architectures of deep learning models are shown in Fig. 4.

The CNN model has been broadly applied to numerous computer vision and NLP applications (*Dahou et al., 2016*; *Severyn & Moschitti, 2015*). For text classification, different word embedding from the sentence or phrase is used. CNN is recognized as more powerful and faster than RNN. Compared to CNN, RNN has less feature consistency (*Yin et al., 2017*). The inputs and outputs of this network are both of a fixed size. When it comes to input and output sizes, RNN is flexible. An RNN is a type of artificial neural network (ANN) that is primarily used in natural language processing and is capable of processing sequential data, recognizing patterns, and predicting the next output. This performs the same operation multiple times on a sequence of inputs. RNN is effective for short-term dependencies (*Can, Ezen-Can & Can, 2018*). The RNN retains information for a short time period after it has been lost. Equations (4) and (5) described the RNN model.

$$h, t = [f(W.[h, (t-1), x, t] + b)] \tag{4}$$

**Table 7** Trainable parameters of deep learning models.

| Models | Trainable parameters |
|---|---|
| GRU | 592,547 |
| RNN | 533,539 |
| BiLSTM | 626,787 |
| LSTM | 624,819 |
| CNN | 567,779 |
| BERT | 109,534,115 |

**Figure 4** Architecture of deep learning models.

where f represents the activation function and W for weights in Matrix, h_(t-1) denotes the hidden state with the preceding time stage, and b indicates the bias-vector.

$$y, t = g(v.h, t + c) \tag{5}$$

In Eq. (5), $y$, $t$ denotes the output with time stage, $g$ is used as an activation function, and $c$ is used as a Bias for output-layers. LSTM (*Rao & Spasojevic, 2016*) is a new variant of RNN that solves the problem of short-term dependencies. The LSTM model is capable of learning long-term dependencies by retaining information for a longer duration. There are three gates in the LSTM; input, output, and forget. LSTM also solves the problem of vanishing gradients. A BiLSTM is composed of two LSTMs: one that takes input in the forward direction and another that takes input in the backward direction. It offers very accurate results for NLP tasks (*Liu & Guo, 2019*). This study used two BiLSTM layers with 64 and 32 units, one 16-bit dense layer, and a final dense layer for classification. The embedding layers used in this study are 5000 ×100 units, followed by dropout layers to avoid overfitting. GRU is less complicated and more efficient because of its reset and update gates. The update gate is responsible for filtering and updating information. In addition, GRU merges cell-state and hidden-state, and its output differs from LSTM's. It overcomes

the vanishing gradient problem as well (*Zulqarnain et al., 2019*). Only five layers are used in the GRU model, which is followed by embedding, GRU, dropout, and two dense layers.

The architectures of deep learning models such as GRU, LSTM, BILSTM, and CNN for smartphone tweets classification are presented in Fig. 4. The embedding layers for these models are used with 50,000 parameters; the purpose of using embedding layers is to convert the input text into the numeric form and during training, each input word is then mapped into vector form, called embedding. The vectors capture the interpretation of input text, and the LSTM layer clearly understands the semantic representation of context.

Dropout layers are used in neural networks for text classification to save the network from overfitting. This layer may be added at any stage in the network and fine-tuned according to the specific tasks. The dropout layer randomly sets neurons to 0 in the training phase and learns the most effective features that work better on unseen data. A dense layer with activation ReLU function is used to provide representations at a high level. The second dense layer works for the final connected layer, also called the classification layer to classify the tweets. The Softmax function is used for multiclass classification with a categorical cross-entropy loss function that differentiates the true and predicted labels. The embedding layer for all the models is used with the same parameters, but other models are used with different layers and parameters with fine-tuning for achieving the highest results.

The BERT (*Singh, Jakhar & Pandey, 2021*) is the most well-known open-source natural language processing model for a variety of applications. BERT is designed to comprehend ambiguous text inside text corpora by associating contexts with textual content. BERT is derived from the deep learning model Transformers. In Transformers, each output element is coupled with each input element, and the weightings between them are dynamically determined by their connection. BERT is often a two-stage model that combines generic pretraining with deep learning.

RNN and CNN are commonly used in language models to solve NLP problems. Even though both RNNs and CNNs are competent models, the Transformer is more competent due to its lack of sequential-processing requirements for input data. Training on more extensive datasets is easy with the help of transformers, which can process input in any order. Therefore, it became much simpler to create pre-trained models like BERT (*Devlin et al., 2018*) which was trained on a large amount of data before its release. Google introduced BERT in 2018 and made it available as open source. During development, the model achieved state-of-the-art results in testing natural language-understanding, including sentiment analysis, semantic role labeling, sentence categorization, and the disambiguation of polysemous words (words with more than one meaning). These accomplishments set BERT apart from previous language models like word2vec and GloVe, which lacked BERT's power to understand the context and polysemous words.

## Proposed BERT model

Figure 5 illustrates the architecture of the proposed BERT model. Three components comprise the BERT model: input word ids, input segment, and input mask. Each 'input id' corresponds to a single subword in the dictionary. The segment tokens are used to identify the segment that each word corresponds to. The input mask is used to distinguish between

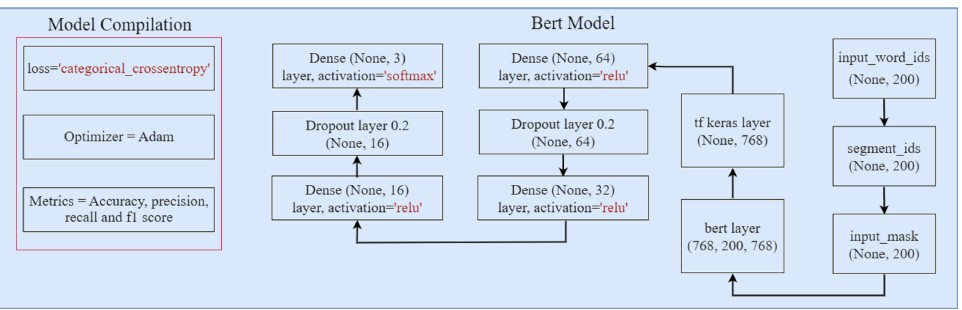

**Figure 5** The architecture of proposed BERT model.

actual-tokens and padding-tokens. To ensure that all input data have the same length, padding-tokens are added to the input. The BERT tokenizer transforms word tokens into input ids and generates the input segment and input mask tensors. In the BERT model, four dense layers are used with 64, 32, 16, and the last dense layer with 3 units for classification. The softmax activation function is used to classify the text. Two 20% dropouts are used to prevent overfitting. To optimize the model's performance, the 'Adam' optimizer and a categorical loss function are used.

## Performance metrics

The performance metric or evaluation metric is commonly used to evaluate the performance of models. The model is trained and tested to evaluate its performance with different metrics like precision, recall, accuracy, and F1 score. Accuracy is calculated by dividing the (true positive + true negative) predictions by the total predictions (true positives+true negatives+false positives+false negatives). The accuracy metric is mostly employed for balanced and imbalanced data, but it cannot perform well with imbalanced data. Another important performance metric for measuring model performance is precision. Determining precision requires dividing true positive predictions by true positives plus false positives. The recall is also known as the capacity to identify all positive instances and missing values, as well as the true positives rate. The recall is calculated by dividing the true positive prediction by the sum of the true positives and false negatives. The F1 score combines the precision and recall score to accurately classify the sentiments.

$$Accuracy = \frac{(TP + TN)}{(TP + TN + FP + FN)} \tag{6}$$

$$Precision = \frac{TP}{(TP + FP)} \tag{7}$$

$$Recall = \frac{TP}{(TP + FN)} \tag{8}$$

$$F1 - score = 2 * \frac{Recall * precision}{Recall + precision}. \tag{9}$$

## RESULTS AND DISCUSSION

This section describes the experimental details for machine learning, deep learning, and transformer-based BERT model. Through hyperparameter optimization and layer modification, 80% of the data is used to train the models. Using the most important preprocessing techniques, unstructured data is cleaned and labeled using the TextBlob methodology. TextBlob calculates subjectivity and polarity based on the cleaned and uncleaned tweets depicted in Figs. 6A and 6B. When the sentiment score is greater than zero, it indicates a positive sentiment; when it is less than zero, it indicates a negative sentiment and when it is equal to zero, it indicates a neutral sentiment. The positive, negative, and neutral sentiments extracted through TextBlob are depicted in Fig. 6C on preprocessed (cleaned) dataset, and Fig. 6D without preprocessed (uncleaned) dataset.

### Results of machine learning models

The results of machine learning are evaluated using four metrics including precision, recall, accuracy, and F1 score. In this experiment, nine models with tuned hyperparameters are utilized. Additionally, the effects of preprocessing techniques are explored. Without using any preprocessing techniques, the SVM attains the highest accuracy of 90%. The KNN model achieves the lowest accuracy of 59% on a dataset of smartphone-related tweets that have not been cleaned or processed. With the preprocessed dataset, the DT achieved 18% higher accuracy than with the raw dataset. Table 8 displays the results of machine learning models with and without preprocessing techniques. The GBM model takes 4890 s for raw data and 988 s for preprocessed data. The SGD model only needs 0.38 s to evaluate preprocessed data. Preprocessing may reduce computational time and effort, as shown in the results. LR obtains an accuracy of 97% with a minimal training time.

The performance of machine learning models is also examined using a dataset of tweets on cryptocurrencies. The models are utilized with the same hyperparameter. In addition, the effects of preprocessing techniques are investigated. SVM achieves the highest accuracy of 90% without any preprocessing technique. The KNN model gets the lowest accuracy of 59% on an uncleaned dataset. The results of machine learning models with and without preprocessing approaches are presented in Table 9.

Figure 7 shows the accuracy of machine learning models with smartphone-related tweets dataset with and without preprocessing. Results indicate that the models perform extremely well with preprocessed datasets. On both preprocessed and without preprocessed datasets, KNN performs poorly. Of the employed models, LR shows the best results with raw and preprocessed datasets and obtains a 97% accuracy when the preprocessed dataset is used for experiments. It is followed by SVM and GBM while KNN shows poor performance.

The impact of preprocessing on computational cost is depicted in Fig. 8. Results demonstrate that time complexity is increased when raw datasets are utilized. When the data is not cleaned, machine learning models require a longer training time. Unclean data contains extraneous information that is not useful. ML models consume less time on a preprocessed dataset, whereas LR takes minimal time and provides the best results. Because the data is not preprocessed, the GBM is quite expensive, taking 4890 s to classify the sentiments.

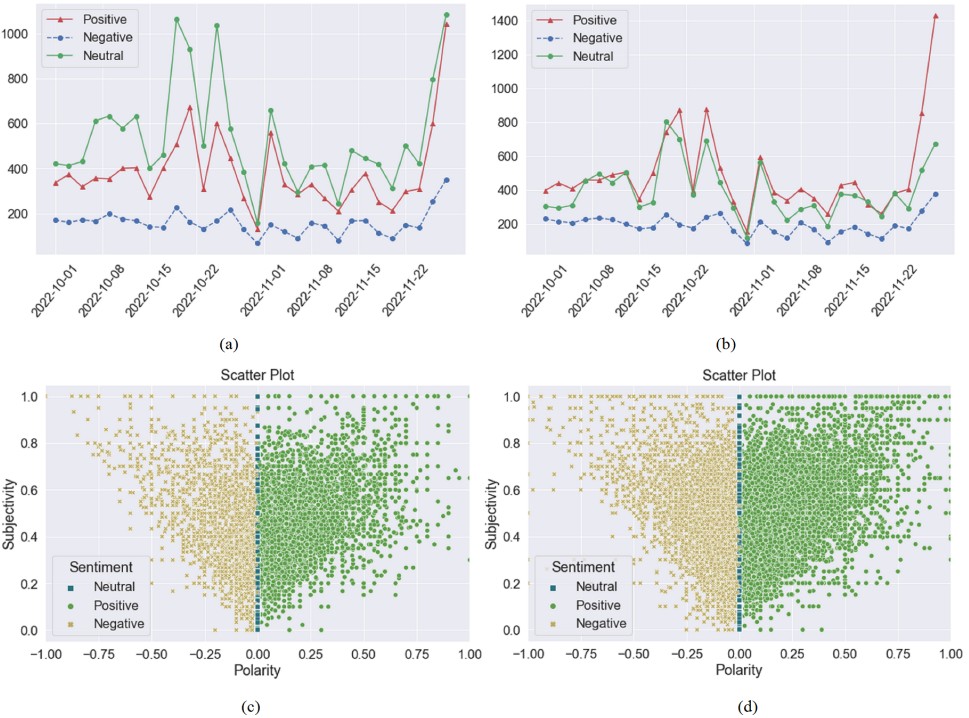

**Figure 6** Positive, negative, and neutral sentiments extracted using the TextBlob technique from October to November are depicted in (A) which describes the sentiments using preprocessing techniques, and (B) which does not use any preprocessing technique. (C) shows the scatter plot of positive, negative, and neutral sentiments with preprocessing, where the $x$-axis shows the polarity score and the $y$-axis shows the subjectivity score, and (D) does so without preprocessing techniques.

**Table 8 Performance of models with preprocessed and without preprocessed dataset.**

| Models | Without preprocessing | | | | | With preprocessing | | | | |
|--------|----------|-----------|--------|----------|------------|----------|-----------|--------|----------|------------|
| | Accuracy | Precision | Recall | F1 score | Time (sec) | Accuracy | Precision | Recall | F1 score | Time (sec) |
| **LR** | 90.052 | 89.981 | 90.052 | 89.958 | 11.146 | 97.252 | 97.239 | 97.259 | 97.230 | 2.897 |
| **RF** | 78.500 | 79.430 | 78.500 | 76.514 | 308.817 | 91.625 | 91.919 | 91.625 | 91.246 | 85.024 |
| **DT** | 78.130 | 77.959 | 78.130 | 77.974 | 22.137 | 96.51 | 96.492 | 96.494 | 96.501 | 6.927 |
| **ETC** | 82.865 | 83.555 | 82.865 | 81.709 | 485.718 | 94.648 | 94.643 | 94.648 | 94.516 | 154.269 |
| **SVM** | 90.083 | 90.154 | 90.083 | 90.105 | 486.559 | 97.362 | 97.356 | 97.357 | 97.356 | 92.699 |
| **SGD** | 88.510 | 88.599 | 88.510 | 88.470 | 0.390 | 96.733 | 96.706 | 96.730 | 96.693 | 0.384 |
| **KNN** | 59.130 | 65.918 | 59.130 | 57.440 | 8.128 | 68.430 | 76.852 | 68.430 | 65.073 | 5.1239 |
| **GBM** | 86.505 | 86.437 | 86.505 | 86.165 | 4890.010 | 96.529 | 96.509 | 96.519 | 96.458 | 988.437 |
| **ADA** | 81.832 | 83.310 | 81.832 | 81.663 | 13.8450 | 88.047 | 88.397 | 89.047 | 87.616 | 7.955 |

## Results of machine learning models using TF-IDF and Word2Vec features

The results of machine learning are also evaluated using TF-IDF and Word2Vec embedding features with evaluation metrics like accuracy, precision, recall, and F1 score. The results with and without preprocessing using TF-IDF features are shown in Table 10. The SVM

**Table 9  Performance of model with preprocessed and without preprocessed crypto-currency dataset.**

| Models | Without Preprocessing | | | | | With Preprocessing | | | | |
|--------|----------|-----------|--------|----------|------------|----------|-----------|--------|----------|------------|
| | Accuracy | Precision | Recall | F1 score | Time (sec) | Accuracy | Precision | Recall | F1 score | Time (sec) |
| **LR** | 89.050 | 88.735 | 89.050 | 88.657 | 1.613 | 93.500 | 93.973 | 93.950 | 93.737 | 0.6404 |
| **RF** | 85.550 | 85.891 | 85.550 | 84.208 | 48.670 | 91.700 | 92.109 | 91.700 | 91.159 | 22.9119 |
| **DT** | 84.650 | 84.038 | 84.651 | 84.220 | 7.032 | 97.150 | 97.122 | 97.150 | 97.129 | 3.340 |
| **ETC** | 87.800 | 87.878 | 86.681 | 87.800 | 74.984 | 94.900 | 94.952 | 94.900 | 94.703 | 41.447 |
| **SVM** | 90.550 | 90.367 | 90.551 | 90.361 | 16.220 | 95.900 | 95.885 | 95.900 | 95.823 | 5.336 |
| **SGD** | 88.350 | 88.076 | 88.352 | 88.061 | 0.124 | 93.950 | 93.905 | 93.803 | 93.950 | 0.132 |
| **KNN** | 70.700 | 75.383 | 70.700 | 65.381 | 0.790 | 77.900 | 81.307 | 77.900 | 73.859 | 0.515 |
| **GBM** | 87.650 | 87.289 | 87.650 | 86.988 | 1197.941 | 97.300 | 97.305 | 97.300 | 97.235 | 529.626 |
| **ADA** | 83.200 | 82.920 | 83.200 | 82.291 | 5.762 | 88.550 | 88.049 | 88.550 | 87.724 | 3.331 |

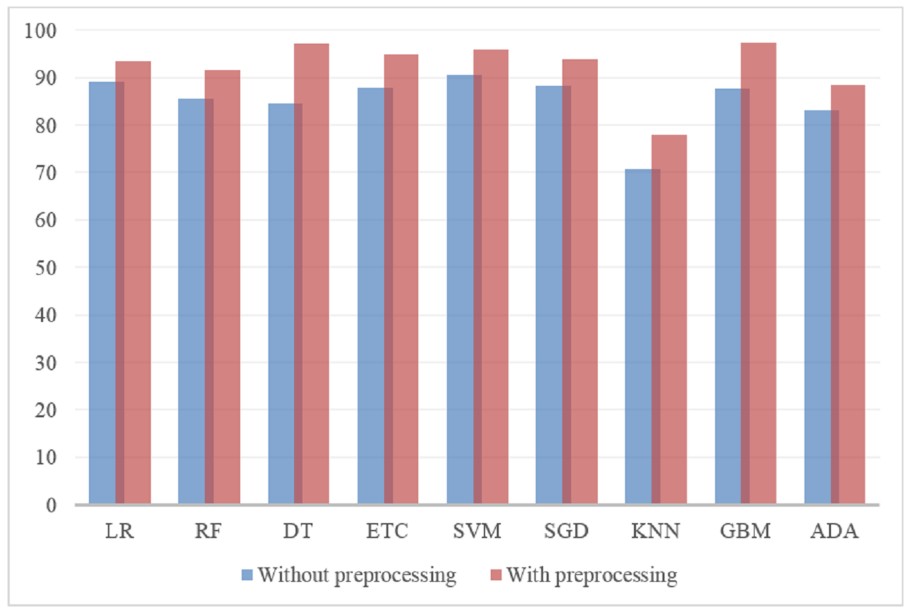

**Figure 7  Results of ML models with preprocessed and without preprocessed datasets.**

attained the highest 97.122% accuracy with preprocessed data and 90.283% without preprocessed data. However, without preprocessing, the SVM model consumes the highest time to process the data. The DT and GBM also perform well with the cleaned dataset, with 96.03% and 95.89% accuracy, respectively. Only SVM reached 90% accuracy with the raw dataset, other models do not perform well and take too much time for training. KNN performs worst with both datasets in terms of preprocessing and without preprocessing. SGD is very efficient concerning time complexity, but the performance is not satisfactory. SGD takes 0.3 s and achieves 93.7% accuracy.

The results with preprocessing and without preprocessing using Word2Vec embedding features are shown in Table 11. The Word2Vec embedding does not capture textual

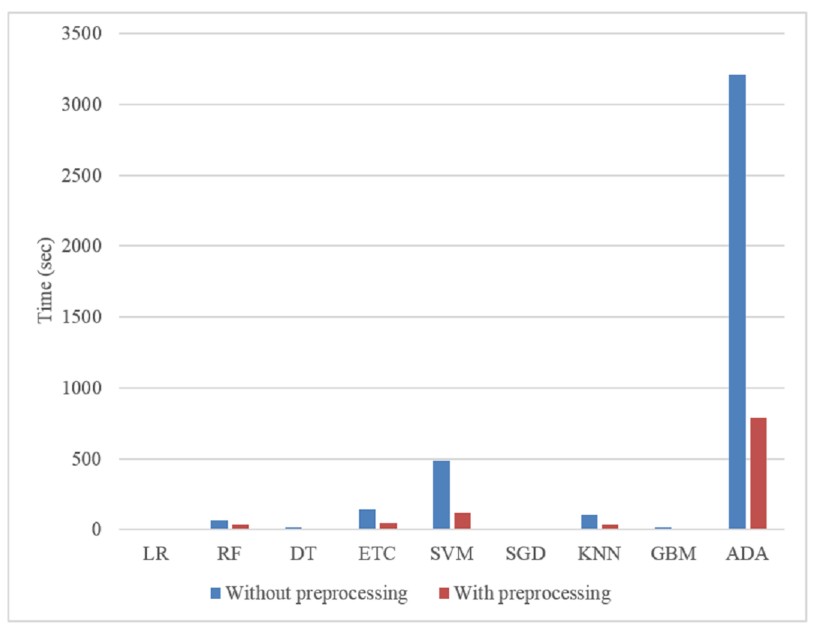

**Figure 8** **Time consumption of ML models on preprocessed and without preprocessed datasets.**

**Table 10** **Performance of models with preprocessed and without preprocessed dataset using TF-IDF features.**

| Models | Without Preprocessing | | | | | With Preprocessing | | | | |
|---|---|---|---|---|---|---|---|---|---|---|
| | Accuracy | Precision | Recall | F1 score | Time (sec) | Accuracy | Precision | Recall | F1 score | Time (sec) |
| LR | 87.446 | 87.346 | 87.446 | 87.167 | 2.756 | 94.744 | 94.856 | 94.744 | 94.639 | 2.814 |
| RF | 75.555 | 77.290 | 75.555 | 73.427 | 68.410 | 91.471 | 91.804 | 91.472 | 91.097 | 32.257 |
| DT | 75.308 | 74.987 | 75.308 | 74.976 | 12.435 | 96.051 | 96.032 | 96.051 | 96.040 | 5.442 |
| ETC | 79.225 | 80.628 | 79.225 | 77.618 | 141.226 | 92.978 | 93.176 | 92.997 | 92.694 | 48.245 |
| SVM | 90.283 | 90.231 | 90.283 | 90.246 | 486.239 | 97.142 | 97.122 | 97.142 | 97.133 | 114.438 |
| SGD | 86.042 | 86.331 | 86.042 | 85.492 | 0.671 | 93.723 | 93.931 | 93.723 | 93.564 | 0.393 |
| KNN | 63.710 | 64.461 | 63.710 | 63.784 | 102.605 | 62.831 | 74.676 | 62.831 | 57.800 | 32.456 |
| ADA | 80.151 | 81.797 | 80.151 | 79.994 | 15.185 | 90.222 | 90.731 | 90.221 | 89.996 | 8.972 |
| GBM | 85.190 | 84.240 | 85.190 | 84.997 | 3210.866 | 95.922 | 95.889 | 95.922 | 95.829 | 791.662 |

information and cannot reproduce context or tone in word-meanings. Also, Word2Vec depends on pre-trained embedding while BoW creates the frequency of each word in the entire document and keeps the out-of-vocabulary words easily. However, machine learning with Word2Vec embedding does not perform well. RF and ETC only attained 71% accuracy with preprocessed data and 65% accuracy without preprocessed data. The SGD model with preprocessing stage attained a very low accuracy of 48% and it takes 0.48 s to classify the sentiments.

## Results of deep learning models

There are many advantages to classifying text using deep learning. It eliminates the need for manual feature engineering and automatically extracts features from unstructured text

**Table 11 Performance of models with preprocessed and without preprocessed dataset using Word2Vec features.**

| Models | Without preprocessing | | | | | With preprocessing | | | | |
|---|---|---|---|---|---|---|---|---|---|---|
| | Accuracy | Precision | Recall | F1 score | Time (sec) | Accuracy | Precision | Recall | F1 score | Time (sec) |
| LR | 60.965 | 59.596 | 60.965 | 58.870 | 12.811 | 64.373 | 64.128 | 64.373 | 61.587 | 8.253 |
| RF | 65.993 | 65.571 | 65.993 | 64.322 | 43.069 | 71.067 | 70.741 | 71.062 | 69.541 | 41.527 |
| DT | 55.798 | 56.182 | 55.709 | 55.964 | 6.758 | 60.749 | 61.177 | 60.749 | 60.944 | 5.568 |
| ETC | 65.515 | 65.066 | 65.515 | 63.868 | 8.178 | 71.169 | 70.706 | 71.169 | 69.769 | 8.654 |
| SVM | 61.181 | 60.908 | 61.181 | 57.364 | 134.504 | 64.065 | 64.594 | 64.065 | 59.687 | 136.955 |
| SGD | 50.370 | 57.689 | 50.370 | 51.517 | 0.783 | 48.264 | 60.728 | 48.264 | 42.901 | 0.486 |
| KNN | 57.248 | 59.273 | 57.248 | 57.875 | 2.432 | 63.485 | 65.990 | 63.448 | 64.405 | 1.991 |
| ADA | 56.215 | 55.036 | 56.215 | 54.828 | 55.1555 | 61.690 | 60.304 | 61.690 | 59.852 | 75.121 |
| GBM | 57.466 | 56.873 | 57.466 | 56.123 | 3031.797 | 62.028 | 61.987 | 62.028 | 63.121 | 720.112 |

input. Deep learning algorithms may be able to identify complex patterns and connections within the data in addition to managing large amounts of data. Unstructured tweets are tokenized and preprocessed. A deep network can then be fed with the tokenized text. The tokenized text must be transformed into numerical-vectors using an embedding layer since deep neural networks work best with numerical data. A word or phrase's meaning and context are captured by an embedding layer, which is a detailed vector representation. The next stage is to use the text data to train a model. The neural networks, having a number of layers and the size of the hidden units, must be set up before data can be passed through the network to determine its weights or biases and evaluated. The performance of deep learning has advanced to the cutting edge in a variety of natural language processing applications. It appears to be a practical method for addressing problems involving text data in the real world.

Table 12 demonstrates that the LSTM deep model achieved 97% accuracy on a preprocessed dataset, whereas the BiLSTM model obtained an accuracy of 88% without preprocessed dataset. Due to gradient-vanishing problems and the high computational cost of the RNN model, which can make it challenging to understand long-term relationships in data, the model was able to achieve 71% accuracy. The proposed BERT-based model achieves the best results with the highest accuracy of 98.57% while the precision, recall, and F1 scores are 98.59%, 98.58%, and 98.58%, respectively. According to the computational time, the RNN model takes 1050 s to process the results on uncleaned data and 540 s on preprocessed data. The CNN model takes very little time 52 s for uncleaned data and 50 s for preprocessed data. It is observed that preprocessing may save computational time and effort. Additionally, it leads to higher performance and makes it easier to process and analyze the data.

Table 13 shows the results of deep learning models for the crypto-currency dataset used for performance validation. It shows that the BiLSTM model performed well on the second dataset, achieving 86% accuracy using the raw dataset, while the LSTM deep model achieved 96% accuracy using the preprocessed dataset. In contrast, the RNN model does not perform well on this dataset. The preprocessing steps applied on unstructured tweets to get cleaned and more structured data help the models to enhance their performance.

**Table 12  Performance of deep learning models with preprocessed and without preprocessed smartphone dataset.**

| Models | Without preprocessing | | | | | With preprocessing | | | | |
|---|---|---|---|---|---|---|---|---|---|---|
| | Accuracy | Precision | Recall | F1 score | Time (sec) | Accuracy | Precision | Recall | F1 score | Time (sec) |
| GRU | 87.615 | 87.557 | 87.615 | 87.578 | 200 | 97.439 | 97.471 | 97.439 | 97.451 | 130 |
| RNN | 71.206 | 70.134 | 71.206 | 70.486 | 1050 | 92.782 | 92.805 | 92.782 | 92.655 | 540 |
| BiLSTM | 88.229 | 88.126 | 88.229 | 88.138 | 320 | 97.577 | 97.581 | 97.577 | 97.576 | 300 |
| LSTM | 87.661 | 87.549 | 87.661 | 87.588 | 190 | 97.856 | 97.870 | 97.856 | 97.860 | 150 |
| CNN | 82.0481 | 81.193 | 82.034 | 81.937 | 52 | 97.316 | 97.331 | 97.316 | 97.322 | 50 |
| **Proposed** | **98.566** | **98.591** | **98.581** | **98.582** | **1216** | **99.349** | **99.352** | **99.350** | **99.351** | **1150** |

Notes.
Bold values indicate the highest values for accuracy, precision, recall and F1 score and shows the superior performance of the proposed appraoch.

**Table 13  Performance of deep learning models with and without preprocessed crypto-currency dataset.**

| Models | Accuracy | Precision | Recall | F1 score | Time (sec) | Accuracy | Precision | Recall | F1 score | Time (sec) |
|---|---|---|---|---|---|---|---|---|---|---|
| GRU | 86.850 | 87.107 | 86.850 | 86.880 | 50 | 96.050 | 96.077 | 96.050 | 96.025 | 50 |
| RNN | 80.650 | 80.092 | 80.650 | 80.354 | 200 | 92.500 | 92.086 | 92.500 | 92.130 | 100 |
| BiLSTM | 87.40 | 87.45 | 87.24 | 87.49 | 100 | 95.640 | 96.603 | 95.640 | 95.613 | 70 |
| LSTM | 86.500 | 86.651 | 78.500 | 86.542 | 50 | 96.250 | 96.231 | 96.250 | 96.235 | 50 |
| CNN | 86.750 | 86.513 | 86.750 | 86.451 | 20 | 94.950 | 94.815 | 94.950 | 94.852 | 15 |
| **Proposed** | **98.000** | **98.000** | **98.00** | **97.991** | **377** | **99.239** | **99.252** | **99.239** | **99.242** | **341** |

Notes.
Bold values indicate the highest values for accuracy, precision, recall and F1 score and shows the superior performance of the proposed appraoch.

The confusion matrix shown in Fig. 9 is used to evaluate the results of classification models. The correct and incorrect predictions made by the model are visualized. The rows represent the true labels, and the columns represent the predicted labels. In the confusion matrix, true positive (TP) refers to observations that are positive and predicted as positive; true negative (TN) to observations that are correctly classified as negative; false positive (FP) is wrongly predicted as positive, and false negative (FN) are the samples incorrectly classified as negative.

The confusion matrix of various machine or deep learning models is shown in Fig. 9, where SVM makes 6,313 correct predictions out of a total of 6,484 predictions with only 171 incorrect predictions. The LR produces 6,306 accurate predictions and 178 inaccurate predictions. The GBM makes 6,259 accurate predictions and 225 inaccurate predictions. From deep learning models, LSTM makes 6,345 accurate and 139 inaccurate predictions. The BILSTM model makes 6,327 accurate predictions and 157 incorrect ones. The GRU model makes 6,318 accurate predictions and 166 incorrect predictions. The proposed BERT model based on Transformers makes just 42 incorrect predictions whereas 6,417 are accurate.

## Comparison of proposed model with state-of-the-art studies

The results of the proposed model are compared with existing state-of-the-art approaches, as given in Table 14. For this purpose, the models from the existing literature are implemented using the dataset used in this study. The SVM model from *Jagdale, Shirsat*

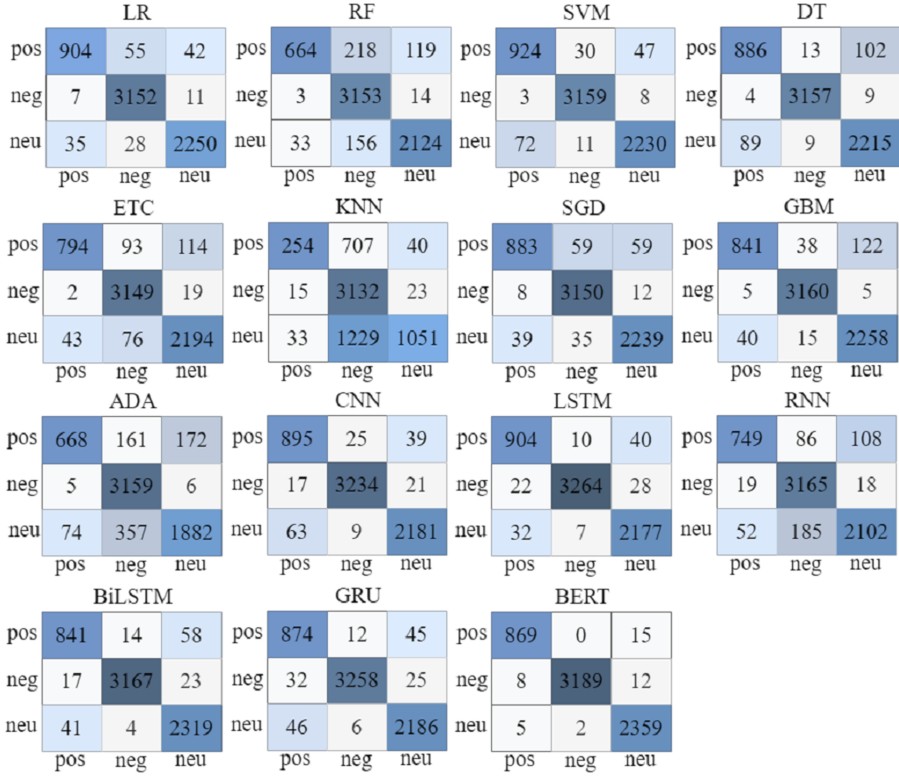

**Figure 9  Confusion matrix for machine learning, deep learning, and proposed BERT model.**

*& Deshmukh (2019)* achieves a 92.85% accuracy. Similarly, a lexicon-based approach from *Chamlertwat et al. (2012)* is used for sentiment classification for the collected dataset. Another SVM model used in *Driyani & Walter Jeyakumar (2021)* achieves an F1 score of 81%. LSTM model from *Iqbal et al. (2022)* can reach 88% accuracy. In addition, SVM model by *Sally (2023)* achieved 76% precision, 60% F1 score, 59% recall score and 78% accuracy on smartphone reviews. The authors do not utilize proper preprocessing approaches and deep learning methods to increase performance. *Yuhan & Huiping (2023)* employed a novel CWSA model using the Chinese smartphone dataset and test the model with only an accuracy metric. Other metrics were not used to evaluate the results, and classification accuracy is not satisfactory. Performance comparison indicates that the proposed BERT model achieved 99% accuracy on the preprocessed dataset.

## Discussions

Sentiment analysis for the top three smartphone brands is performed in this study. Unstructured and unlabeled tweets are collected from Twitter in this regard. Various preprocessing steps are applied to transform the raw text into a more structured and clean text. The relevant features are extracted from tweets using BoW. It is challenging to identify the most significant smartphone brand because it depends on individual opinions and requirements. Sentiment analysis helps in determining people's attitudes toward their favorite smartphone brands. The analysis of sentiments shows that between Apple and

**Table 14   Comparison with state-of-the art studies.**

| Paper Ref. | Approach/Method | Datasets | Accuracy | Precision | Recall | F1 score |
|---|---|---|---|---|---|---|
| *Jagdale, Shirsat & Deshmukh (2019)* | SVM | Amazon product reviews | 92.85% | 91.64% | – | 95.64% |
| *Chamlertwat et al. (2012)* | Lexicon based approach | Smartphone brands tweets | The authors only extract positive or negative sentiments. | | | |
| *Fang & Zhan (2015)* | SVM, Naïve Bayes, RF | Amazon product reviews | – | – | – | 81% |
| *Driyani & Walter Jeyakumar (2021)* | SVM with RBF kernel | Apple iPhone reviews | 91.87%. | – | – | – |
| *Dhabekar & Patil (2021)* | LSTM | Amazon products | 93% | 93% | 93% | 92% |
| *Iqbal et al. (2022)* | LSTM | Cell Phone | 88% | 98% | 64% | 70% |
| *Sally (2023)* | SVM | Smartphone Reviews | 78% | 76/% | 59% | 60% |
| *Supriyadi & Sibaroni (2023)* | Indo-BERT | Xiaomi Smartphone tweets | 90% | – | – | – |
| *Yuhan & Huiping (2023)* | CWSA model | Chinese Smartphone tweets | – | – | – | 89.6% |
| This study | BERT | Smartphone | 99.34% | 99.35% | 99.35% | 99.35% |

Samsung, most people prefer Apple smartphones over Samsung. 40% of people favor Apple smartphones, while 32% prefer Samsung mobile phones. Only 13% of people dislike Apple smartphones while 16% dislike Samsung.

This study investigates the results of various machine and deep learning models with various layers and also analyzes the effect of various preprocessing techniques. The experiments show that the SVM machine learning model classifies the smartphone-related sentiments with an accuracy of 90% without any preprocessing. DT achieves 18% higher accuracy with preprocessed data as compared to raw data. Also, without preprocessing, models take longer time, as GBM takes 4849 s for raw data and only 988 s for preprocessed data. From deep learning models, the RNN model takes the highest computational time for unprocessed data as compared to preprocessed data. The LSTM model achieved a 97% accuracy with preprocessed data. The proposed Tranformer-based BERT model achieved the highest 99% accuracy. As seen in Tables 8 and 12, preprocessing textual data is essential for NLP tasks because it enables better model training and obtains better results. The models show poor performance from the text data if it is not properly cleaned and standardized.

## Limitations

We performed sentiment classification using 32K smartphone-related tweets using nine fine-tuned and well-known machine learning as well as deep learning models. We also proposed Transformer based deep model to accurately classify the tweets with excellent results. However, there are some limitations to deep and transformer-based models for text classification. Deep learning requires large datasets for training. With limited datasets, deep learning performs well on the training set, but when we test on unseen tweets, their performance decreases and leads to overfitting. Tweets are user-generated texts that may be noisy and lack contextual information. In the future, we intend to collect a large dataset from various social platforms such as Facebook, Instagram, *etc.*, other than Twitter with unique smartphone brand aspects like price, camera quality, and operating system

and perform text analysis with other well-known feature engineering techniques and transformers.

## CONCLUSIONS

This study presents a BERT-based model for the classification of sentiments related to smartphones. The model is evaluated using a self-collected tweets dataset and validated on an additional dataset. In addition, the influence of preprocessing is evaluated on time complexity and the model's performance. Results indicate that the proposed approach is able to obtain a 99% accuracy using the preprocessed data. Performance comparison with nine well-known machine learning models and deep learning models including GRU, RNN, BiLSTM, LSTM, and CNN show that the proposed approach outperforms these models. Similarly, the proposed approach shows better results than existing state-of-the-art approaches. Results demonstrate that the use of preprocessing steps improves models' performance and reduces computational time. This study utilizes BoW, TF-IDF, and Word2Vec embedding approaches for feature extraction, while other well-known features are left for the future. The SVM achieved 97.4% accuracy using BoW features using preprocessed data, and the LR achieved 97.3% accuracy. While utilizing the TF-IDF and Word2Vec features, SVM only achieved 97.1% and 64.01% accuracy score, respectively. The results proved that BoW extracted more important features from smartphone related tweets than TF-IDF and Word2Vec. The performance of machine learning with Word2Vec features is very low. In addition, the proposed BERT transformer, which is often a two-stage model that combines generic pretraining with deep learning, achieved excellent results on both self-collected and Kaggle datasets. The proposed methodology can be implemented on other social media related sentiments analysis tasks and can be very beneficial for companies to get insights from the sentiments and better the brands. We also intend to apply other preprocessing techniques and analyze their impact on the model's performance.

### Funding
The authors received no funding for this work.

### Competing Interests
Imran Ashraf is an Academic Editor for PeerJ Computer Science. Sudheesh R is employed by Kodiyattu Veedu

### Author Contributions
- Sudheesh R conceived and designed the experiments, analyzed the data, prepared figures and/or tables, and approved the final draft.
- Muhammad Mujahid conceived and designed the experiments, analyzed the data, prepared figures and/or tables, and approved the final draft.
- Furqan Rustam conceived and designed the experiments, analyzed the data, prepared figures and/or tables, and approved the final draft.

- Bhargav Mallampati performed the experiments, analyzed the data, prepared figures and/or tables, and approved the final draft.
- Venkata Chunduri performed the experiments, performed the computation work, authored or reviewed drafts of the article, and approved the final draft.
- Isabel de la Torre Díez performed the experiments, performed the computation work, authored or reviewed drafts of the article, and approved the final draft.
- Imran Ashraf performed the experiments, performed the computation work, authored or reviewed drafts of the article, and approved the final draft.

## Data Availability

The code and data are available in the Supplemental Files.

## Supplemental Information

Supplemental information for this article can be found online at http://dx.doi.org/10.7717/peerj-cs.1432#supplemental-information.

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
