# Peer review of "Bidirectional encoder representations from transformers and deep learning model for analyzing smartphone-related tweets"

_PeerJ Computer Science, doi:10.7717/peerj-cs.1432_

## Round 0.1 · original submission · Minor Revisions

Dear authors,

Your article has not been recommended for publication in its current form. However, we do encourage you to address the concerns and criticisms of the reviewers and resubmit your article once you have updated it accordingly.

Reviewer 1 ·

Basic reporting

First of all, I would like to congratulate the author(s) for their work. Please be assured that I will evaluate the study only from a scientific point of view, without prejudice.

This work is a valuable and prudent work in the computer science community and social media area. Also, I can't end without saying that it is worth reading.

In the study, the author(s) evaluated by using Machine learning and deep learning, and transformer-based BERT approach models to predict sentiment analysis.

In addition, the authors claim that the proposed model shows superior performance compared to the classical methods.

Experimental design

Design and Writing

The article should be prepared by considering the journal template writing rules (see: https://peerj.com/about/author-instructions/cs).

Although the article was prepared with great effort, there are typographical errors in the most prominent parts of the article.

These deficiencies can't be tolerated for the valuable study.
From the use of abbreviations to reference notation, the article should be completely revised.

ex1. It is a simple feature extraction approach, yet produces good results.

- Also, choose correctly the verbs that emphasize the achievement results of the study.

ex2: The study Iqbal et al. (2022) used a deep learning long short-term memory (LSTM) model with various combinations of layers for sentiment categorization on five different product review datasets obtained from Twitter and Amazon.

- Attention should be paid to the use of abbreviations.

ex3: Sentiment classification is a major area of research in natural language processing (NLP), and various studies have utilized it to determine sentiments regarding different products and services.

- Some of your sentences should be shortened and clarified.

ex4. Several studies
just collected tweets and classified them into positive or negative sentiments, while many others deployed machine learning and deep learning models.

- Incomplete and inconsistent sentences should be completed. In addition, the article should be cleared of all grammatical errors by using the grammar check tool (For example, Grammerly or Ginger).

Validity of the findings

In the abstract section, clearly emphasize your main motivation for the preparation of the article.

Also, express numerically all the performance metrics obtained by the method proposed in the summary section.

The last sentence of the Abstract section must have been completed with a more striking sentence.

In the conclusion section, list your achievements with this study.

Clearly express your contribution to future studies and literature.

Compare your performance metrics with previous studies in the literature.

Satisfactory technical information about the method used is not presented. No mathematical expressions, from the evaluation metrics used to the algorithms, are included in the article.

Please clearly describe the methods and algorithms used.

Specify the parameters of the algorithms used separately.

Provide the motivation for the conclusion part.

Additional comments

In addition, in order to examine the latest technology artificial intelligence and optimization algorithms, read the following articles and add all that seem relevant to your article.

2022, Deep-Cov19-Hate: A Textual-Based Novel Approach for Automatic Detection of Hate Speech in Online Social Networks throughout COVID-19 with Shallow and Deep Learning Models

2021, Performance Assessment of Artificial Intelligence-Based Algorithms for Hate Speech Detection in Online Social Networks https://doi.org/10.35234/fumbd.986500

2021, Metaheuristic Ant Lion and Moth Flame Optimization based Novel Approach for Automatic Detection of Hate Speech in Online Social Networks

2021, Sentiment Analysis in Social Networks Using Social Spider Optimization Algorithm

Reviewer 2 ·

Basic reporting

1-Authors should review the article by paying attention to some spelling and spelling rules.
2-It is seen that many studies in the literature are not included. Related studies should be developed.

Experimental design

1- It is very important that the BERT model achieves higher performance than methods without a preprocessing stage such as deep learning.
2-I agree that the features selected with the Bag of Words technique achieve very high classification accuracy. However, why TF-IDF and Word Embeddings methods, which are feature extraction methods, are not considered in the study.
3-The limitations of the study should be discussed extensively in the discussion section.

Validity of the findings

1-The conclusion part of the study should be developed. The importance of the work and the success achieved should be emphasized.

---

## Round 0.2 · accepted · Accept

Dear authors,

Thank you for clearly addressing all of the reviewers' comments. Your article is accepted for publication now.

Best wishes,

Reviewer 1 ·

Basic reporting

I would like to congratulate the author(s) who contributed to the preparation of the article.

I have seen that the revision file was prepared with great effort.

In this section, the desired deficiencies are completely completed.

I suggest rechecking any typos in the article.

Experimental design

I congratulate the author(s) who contributed to the preparation of the Experimental design section of the article.

I saw that the revision file was prepared with great effort.

In this section, the desired deficiencies are completely completed.

I recommend that you double-check the accuracy of the tables and figures in the article.

Validity of the findings

With this revision, I can say that the article has become more readable and understandable as it is.

All desired changes have been completed in this section.

I congratulate the author(s) who contributed to the preparation of the Validity of the findings section of the article.

Additional comments

Considering the additional comments, the necessary revision has been completed.

With this revision version, I can state that the article has become more readable and powerful with this version.

I congratulate the author(s) who contributed to the preparation of the comments section of the article.

Reviewer 2 ·

Basic reporting

The authors have fulfilled all our demands in the article.

Experimental design

The authors have fulfilled all our demands in the article.

Validity of the findings

The authors have fulfilled all our demands in the article.

Additional comments

The authors have fulfilled all our demands in the article.